# On the Training Dynamics of Contrastive Learning with Imbalanced Feature Distributions: A Theoretical Study of Feature Learning

**Haixu Liao**
Department of Data Science
New Jersey Institute of Technology
Newark, NJ 07102
hl534@njit.edu

**Yating Zhou**
Department of EECS
Rensselaer Polytechnic Institute
Troy, NY 12180
zhouy26@rpi.edu

**Songyang Zhang**
Department of Electrical and Computer Engineering
University of Louisiana at Lafayette
Lafayette, LA 70504
songyang.zhang@louisiana.edu

**Shuai Zhang**
Department of Data Science
New Jersey Institute of Technology
Newark, NJ 07102
shuai.zhang@njit.edu

## Abstract

Contrastive learning has served as a powerful framework in the early development of vision–language models (VLMs), demonstrating remarkable effectiveness in learning generalizable representations and establishing itself as the foundation for many state-of-the-art systems. However, despite these advances, its theoretical understanding remains limited, particularly under imbalanced data distributions that are prevalent in real-world settings. Such imbalance can degrade representation quality and induce biased model behavior, yet a rigorous characterization of these effects is still lacking. In this work, we develop a theoretical framework to analyze the training dynamics of contrastive learning with Transformer-based encoders under imbalanced data. Our results reveal that neuron weights evolve differently across three stages of training, with distinct dynamics for majority features, minority features, and the noise. We further show that minority features diminish neurons' representational capacity, increase the need for more complex architectures, and impair the separation of ground-truth features from noise. These findings offer new theoretical insights into how data imbalance shapes learning in contrastive frameworks and serve as an early step towards principled modifications for developing more robust and unbiased representations.

## 1 Introduction

Contrastive learning has emerged as a powerful paradigm in representation learning, effectively leveraging unlabeled data without relying on human-annotated labels. Within this framework, samples with similar semantic meaning are treated as positive pairs, while those with different semantics are considered negative pairs. By pulling positive pairs closer together and pushing negative pairs farther apart in the representation space, contrastive learning enables models to capture rich and discriminative features. Compared with supervised learning, the resulting representations are often more robust and less sensitive to noise [18, 4, 19, 8]. This approach has demonstrated remarkable success across a wide range of applications and has been particularly influential in multi-modal learning [11, 9], driving major advances in the early development of vision-language models [13].

Despite its strengths, contrastive learning faces challenges when applied to real-world datasets with class imbalance. In such scenarios, majority classes dominate the sample space, while minority classes with limited samples are underrepresented in both positive and negative pair formation. This imbalance can hinder the learning process, causing the model to under-capture discriminative features associated with minority classes, ultimately degrading representation quality. Several studies have attempted to address the challenge of contrastive learning under imbalanced data. One line of research focuses on sample re-weighting strategies, which aim to balance the influence of minority and majority class samples [1, 6, 10, 16, 14]. Another line of work explores data resampling methods, such as oversampling minority data or undersampling majority data, to achieve a more balanced training distribution [2, 5, 12, 15]. However, both approaches rely heavily on accurate estimation of re-weighting or re-sampling ratios, which is an aspect that is often difficult to characterize precisely and typically depends on human intuition or heuristic methods.

Despite the progress made by these approaches, most efforts have been largely empirical, relying on heuristic methods to alleviate the imbalance problem. While these techniques often provide performance gains in practice, they do not explain why or how imbalance undermines the quality of learned representations. Recent work has begun to develop theoretical understandings of contrastive learning, primarily addressing questions such as its superiority over traditional generative approaches like GANs [7], the necessity of data augmentation for effective representation learning [17], and its ability to produce representations that reduce the sample complexity of downstream tasks [3]. Nonetheless, these studies have not considered the implications of imbalanced data distributions.

Most existing studies on contrastive learning focus on empirical performance, while its theoretical foundations, especially for feature learning, remain less understood. In this work, we provide a theoretical analysis of how neurons learn feature representations through contrastive training. We study a simplified but representative setting: a Transformer-MLP framework with a single-head attention mechanism followed by an MLP with bilateral ReLU activations. To make the analysis clear, we use a structured data model where each input includes majority and minority features with different frequencies. This setup highlights the key role of feature frequencies and helps us describe their impact on training dynamics and how neurons learn features. In turn, the model allows us to formalize how contrastive learning enhances majority features and drives neurons to learn purer feature representations. Overall, our paper makes two main **contributions**:

**First, we develop a theoretical framework to characterize the training dynamics of contrastive learning under Transformer-based encoders with an imbalanced data distribution.** Our results show that neuron weights evolve differently when learning majority features, minority features, and noise across the three stages of training.

**Second, we quantitatively characterize how the presence of minority features influences neurons' learning capacity and, consequently, representation learning.** Our analysis shows that neurons learn majority features more quickly, while minority features are acquired more slowly. Moreover, in the presence of minority features, capturing effective representations requires a more complex neural network, and the neurons' ability to distinguish ground-truth features from noise becomes degraded.

## 2 Problem Formulation

**Contrastive Learning Framework.** Let $X = [x^{(1)}, \ldots, x^{(L)}] \in \mathbb{R}^{d_1 \times L}$ be an input sequence with $L$ tokens. The goal of contrastive learning is to learn a mapping $h(\cdot) : \mathbb{R}^{d_1 \times L} \to \mathbb{R}^m$ that outputs a meaningful embedding from the input sequence.

Let $(X_n, X_{n'})$ denote a *positive pair* (e.g., derived from the same objective or sharing semantic meaning), and let $\mathfrak{N}$ denote a set of corresponding *negative samples* (e.g., random samples). The InfoNCE loss with temperature parameter $\tau > 0$ is defined as:

$$\ell(f, X_n, X_{n'}, \mathfrak{N}) := -\log \left( \frac{e^{\text{sim}_f(X_n, X_{n'})/\tau}}{\sum_{x \in \{X_{n'}\} \cup \mathfrak{N}} e^{\text{sim}_f(X_n, x)/\tau}} \right), \tag{1}$$

where the similarity function is given by $\text{sim}_f(X_n, X_{n'}) := \langle f(X_n), \text{StopGrad}(f(X_{n'})) \rangle$, and $\text{StopGrad}(\cdot)$ acts as the identity in forward pass while blocking gradients in backpropagation.

Then, the learning objective is to minimize an empirical risk with $\ell_2$-regularizer over a batch of size $K$, i.e.,

$$\widehat{L}_{\text{aug}}(f_t) = \widehat{L}(f_t) + \frac{\lambda}{2}\|w\|_F^2 = \frac{1}{K}\sum_{k=1}^{K}\ell(f; X_k, X_{k'}, \mathfrak{N}^k) + \frac{\lambda}{2}\|w\|_F^2, \qquad (2)$$

where $w$ is the neural network parameters.

**Model Architecture: Transformer-MLP.** We employ a simplified single-head self-attention mechanism on top of an MLP layer. Each input sequence is passed through the attention layer, where every token serves as a query. Then, it is followed by a bilateral ReLU (BReLU) activation in the MLP layer, where $\text{BReLU}_b(s) = \text{ReLU}(s - b) - \text{ReLU}(-s - b)$. Specifically, the embedding function $f$ is expressed as

$$f(X_n) = \big(h_1(X_n), \dots, h_m(X_n)\big)^\top \in \mathbb{R}^m,$$

$$\text{with} \quad h_i(X_n) = \sum_{r=1}^{L} \text{BReLU}_{b_i^{(t)}}\Big(\langle w_i^{(t)}, \text{Attention}(W_Q x_n^{(r)}, W_K X_n, W_V X_n)\rangle\Big). \qquad (3)$$

In this early stage of our analysis, we fix attention layer weights to identity matrices and focus on the MLP layer weights. Note that the analysis of this model still differs substantially from a standard feedforward network because the preceding self-attention aggregates information across tokens.

## 3 Theoretical Analysis

### 3.1 Key Insights of the Findings

**(K1). Training dynamics of contrastive learning based on the Transformer-MLP framework.** The theory shows that the learning process can be divided into three stages. In the first stage, neuron weights in feature directions start to increase, while their components in non-feature directions stay almost unchanged. In the second stage, the alignment with feature directions keeps growing, and the learned features become purer, while non-feature directions remain suppressed. In the final stage, each neuron aligns with a specific set of features $\mathcal{N}_i$, on which it already had some degree of alignment at initialization.

**(K2). Theoretical characterizations of how imbalanced data in affecting the neuron's learning ability.** In the first stage of training, neurons start to increase along feature directions, and the speed of this growth depends on the feature frequency $\epsilon_j$. Features with larger $\epsilon_j$ grow faster, so neurons align with them more quickly. Features with smaller $\epsilon_j$ grow more slowly, and neurons may need more time to capture them. In the second stage, this difference becomes more visible, as neurons that follow features with larger $\epsilon_j$ keep increasing their alignment, while features with smaller $\epsilon_j$ continue to evolve at a slower pace.

**(K3). The effect of the ratio $\epsilon_{\min}/\epsilon_{\max}$ on the final learning state.** In the final stage of training, the feature frequency ratio $\epsilon_{\min}/\epsilon_{\max}$ controls how neurons distribute their weights across different features. For minority features, $\epsilon_j = \epsilon_{\min}$, so the ratio directly determines the size of the coefficient $\alpha_{i,j}$. When the ratio $\epsilon_{\min}/\epsilon_{\max}$ is small, each $\alpha_{i,j}$ for minority features becomes very small. As a result, the set $\mathcal{N}_i$ becomes larger, meaning that each neuron aligns with more features that had some degree of initialization alignment. However, the contribution from each feature is weaker, so the neuron ends up mixing many features together in a more mixed way. In contrast, when the ratio $\epsilon_{\min}/\epsilon_{\max}$ is larger, the coefficients $\alpha_{i,j}$ become stronger. In this case, the set $\mathcal{N}_i$ becomes smaller, and each neuron aligns with fewer features. This makes the final representation more concentrated, and the features learned by each neuron are purer.

### 3.2 Formal Theoretical Results

Theorem 3.1 describes two main effects of gradient descent in the first stage of training: (i) In the feature directions, the neuron weights increase rapidly as shown in (4), while in the non-feature directions they are suppressed as shown in (5) during training. (ii) The growth of a neuron's weight in a feature direction $\mathbf{M}_j$ depends on the frequency $\epsilon_j$ as shown in (4). Larger $\epsilon_j$ leads to faster growth, while smaller $\epsilon_j$ results in slower growth, making the feature harder to capture in the early stages of training. In short, the feature frequency $\epsilon_j$ directly controls how much the inner product $\langle w_i^{(t)}, \mathbf{M}_j\rangle$ increases under gradient descent.

Table 1: Summary of main notations

| | | | |
|---|---|---|---|
| $\eta$ | Learning rate | $\lambda$ | Regularization parameter |
| $\tau$ | Temperature coefficient | $K$ | Batch size |
| $w_i^{(t)}$ | The Neuron $i$ after $t$ steps of GD | $\mathbf{M}_j$ | The feature vector of feature $j$ |
| $\mathfrak{N}$ | Set of negative samples | $\mathfrak{B}$ | The set of $X_{n'}$ and negative samples |
| $\epsilon_{\min}$ | frequency of minority feature | $\epsilon_{\max}$ | frequency of majority feature |
| $\epsilon_j$ | Feature frequency for feature $j$ | $\mathcal{N}_i$ | Set of features for ordinary neuron $i$ |
| $\mathcal{M}_j$ | Set of ordinary neurons for feature $j$ | $\mathcal{M}_j^\star$ | Set of lucky neurons for feature $j$ |

**Theorem 3.1** (**Stage 1**). *During the first training stage, the update of neuron weights $w_i^{(t)}$ at time $T_1$ can be bounded as follows.*

$$|\langle w_i^{(T_1)}, \mathbf{M}_j\rangle| \geq |\langle w_i^{(0)}, \mathbf{M}_j\rangle|(1 + \epsilon_j C_z \log d) - \tilde{O}\Big(\frac{\|w_i^{(T_1)}\|_2}{\mathrm{poly}(d)}\Big) \tag{4}$$

$$|\langle w_i^{(T_1)}, \mathbf{M}_j^\perp\rangle| \leq |\langle w_i^{(0)}, \mathbf{M}_j^\perp\rangle| + \tilde{O}\Big(\frac{\|w_i^{(T_1)}\|_2}{\mathrm{poly}(d)}\Big) \tag{5}$$

Theorem 3.2 describes the gradient descent dynamics in the second stage of training, focusing on how neurons behave in different directions. (i) For neurons that belong to $\mathcal{M}_j^\star$, their inner product with the feature vector keeps increasing as shown in (6) (ii) In contrast, along the noise direction ($\mathbf{M}_j^\perp$), the growth stays almost unchanged as shown in (7). In particular, the value of $\epsilon_j$ determines how quickly neurons in the feature directions evolve during training.

**Theorem 3.2** (**Stage 2**). *During the second training stage, the update of neuron weights $w_i^{(t)}$ at time $T_2$ can be bounded as follows. For each $j \in [d]$, if $i \in \mathcal{M}_j^\star$, then:*

$$|\langle w_i^{(T_2)}, \mathbf{M}_j\rangle| \geq \Omega(1)\|w_i^{(T_2)}\|_2 \tag{6}$$

*If along the orthogonal non-feature direction $\mathbf{M}_j^\perp$:*

$$|\langle w_i^{(T_2)}, \mathbf{M}_j^\perp\rangle| \leq |\langle w_i^{(T_1)}, \mathbf{M}_j^\perp\rangle| + \tilde{O}\Big(\frac{\|w_i^{(T_2)}\|_2}{\mathrm{poly}(d)}\Big). \tag{7}$$

Theorem 3.3 describes the feature learning behavior of neurons in the final stage. Specifically, we prove that: (i) Each neuron weight vector $w_i$ eventually aligns with a set of features $\mathcal{N}_i$. This set corresponds to the features that already had some degree of alignment with $w_i$ at initialization. (ii) The size of $\mathcal{N}_i$ depends on the ratio $\epsilon_{\min}/\epsilon_{\max}$. A smaller ratio enlarges $|\mathcal{N}_i|$, leading neurons to encode more mixed features, whereas a ratio closer to one yields smaller $|\mathcal{N}_i|$, so neurons capture purer features that benefit representation learning. (iii) For each feature $\mathbf{M}_j$, the number of neurons that contain some degree of initialization component along $\mathbf{M}_j$ admits an upper bound. Moreover, there are at least $\Omega(d^{\omega_1})$ neurons with $\mathcal{N}_i = \{j\}$, where $\omega_1 = C_m - \left(\frac{\epsilon_{\max}}{\epsilon_{\min}}\right)^2 (1 + \gamma c_0)$, indicating imbalanced data leads to less number of neurons in learning the purified feature.

**Theorem 3.3** (**Stage 3: Neuron–Feature Alignment in Contrastive Learning**). *Let $m = d^{C_m}$ be the number of neurons and $\tau = \mathrm{polylog}(d)$. Suppose we train the neural net $f_t$ via contrastive learning, and consider iterations $T \in [T_3, T_4]$ with $T_3 = \frac{d^{1.01}}{\eta}$ and $T_4 = \frac{d^{1.99}}{\eta}$. Then the following guarantees hold:*

$$\frac{1}{T}\sum_{t\in[T]} L_{\mathrm{aug}}(f_t) \leq o(1), \quad \frac{1}{T}\sum_{t\in[T]} \mathcal{L}(f_t) \leq o(1). \tag{8}$$

*Moreover, for each neuron $i \in [m]$ and $t \in [T_3, T_4]$, the weight will learn the following set of features:*

$$w_i^{(t)} = \sum_{j\in\mathcal{N}_i} \alpha_{i,j}\mathbf{M}_j + \sum_{j\notin\mathcal{N}_i} \alpha'_{i,j}\mathbf{M}_j + \sum_{j\in[d_1]\backslash[d]} \beta_{i,j}\mathbf{M}_j^\perp, \tag{9}$$

where $\alpha_{i,j} \in \left[\frac{\epsilon_j}{\epsilon_{\max}}\frac{\tau}{\Xi_2}, \frac{\epsilon_j}{\epsilon_{\max}}\tau\right], \alpha'_{i,j} \leq o\left(\frac{\epsilon_j}{\epsilon_{\max}}\frac{1}{\sqrt{d}}\right)\|w_i^{(t)}\|_2, |\beta_{i,j}| \leq o\left(\frac{1}{\sqrt{d_1}}\right)\|w_i^{(t)}\|_2$. *Furthermore, the size of $\mathcal{N}_i$ is bounded by $|\mathcal{N}_i| = O\left(d^{1-\left(\frac{\epsilon_{\min}}{\epsilon_{\max}}\right)^2 \cdot (1-\gamma c_0)}\right)$. Finally, for each dictionary atom $\mathbf{M}_j$, there are at least $\Omega(d^{\omega_1})$ neurons $i \in [m]$ such that $\mathcal{N}_i = \{j\}$.*

## 4 Numerical Experiments

Following our learning setup, we validate our theoretical insights on synthetic data with parameters $m = 30$ and $d = 9$ (Details can be found in supplementary materials). In Figure 1, the x-axis represents the feature index, and the y-axis represents the neuron index, where we only plot the first 13 neurons to save space. Each entry corresponds to the projection of a neuron's weight onto the direction of the associated feature; larger values indicate stronger alignment between the neuron and that feature. The first five features (columns 1–5) are majority features, while the last four (columns 6–9) are minority features. As the figure illustrates, neurons exhibit significantly larger projections onto majority features. Nearly every neuron is strongly associated with at least one majority feature. At the same time, each majority feature is represented by at least one neuron, and in such cases, the projection onto that feature is substantially larger than onto all others, meaning the feature dominates the neuron's representation. This demonstrates that majority features are easier to learn and tend to be represented by multiple neurons, in contrast to the minority features.

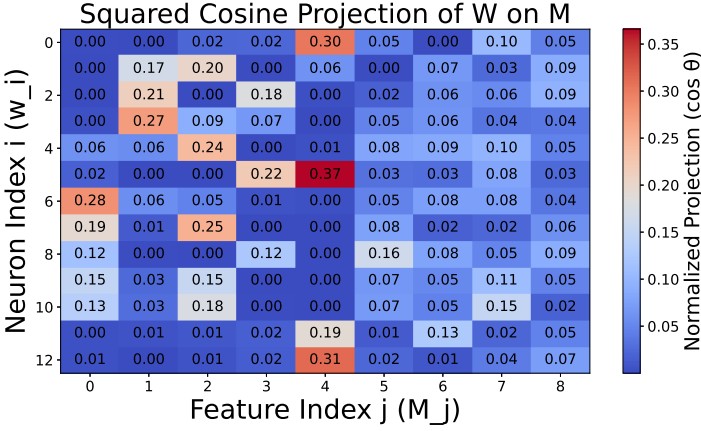

Figure 1: Squared cosine projection of the first 13 neurons ($w_i$) on 9 dictionary atoms ($M_j$). The first five atoms are majority features, and the last four are minority features.

## 5 Conclusion

This work takes a step toward a principled understanding of how imbalanced data shapes the dynamics of contrastive learning in Transformer-based encoders. Our analysis shows that imbalance harms performance: minority features reduce neurons' representational capacity, increase the demand for more complex architectures, and hinder the separation of ground-truth features from noise. Looking ahead, a promising direction is to investigate how these insights can inspire the design of more principled methods or help explain the effectiveness of existing approaches in addressing imbalance in contrastive learning.

## Acknowledgments

This work was supported by the National Science Foundation (NSF) #2349879 and #2349878. We also thank all anonymous reviewers for their constructive comments.

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
