# A  Related Works

**Convergence and Generalization Analysis of Contrastive Learning:** Despite its empirical success, contrastive learning lacks a mature theoretical understanding, largely due to the complexity of its loss function. Early research investigates why augmentation is essential for the success of contrastive learning, showing that such an alignment between augmented positive pairs facilitates learning useful representations [28, 35, 29, 39, 33, 4]. [34, 37] establishes a connection between the gradients of contrastive learning and graph neural networks, highlighting interpretability through a graph-theoretic perspective. [10] also explores the connections between contrastive learning and graph theory, proposing a new loss function linked to graph spectral clustering to help explain its success. [39] emphasizes the necessity of data augmentation for breaking dependencies on spurious noise. None of these works has explored how imbalanced data influences the training dynamics of contrastive learning.

**Feature Learning Paradigm:** The mathematical framework in this paper is closely related to the feature learning paradigm. Specifically, we assume the data follow a sparse coding model, which is a mixture of latent features, and study the training dynamics of model weights to examine how they align with these features. Most prior works focus on supervised learning [1, 42, 18, 3], where features are tied to ground-truth labels; however, such settings cannot be directly extended to contrastive learning. Due to the complexity of analyzing fine-grained training dynamics, existing studies are typically limited to simple one-hidden-layer neural networks, with some recent efforts exploring Transformers but still restricted to a single layer [13, 24, 17], even in supervised settings. The most relevant work is [39], which analyzes the training dynamics of contrastive learning with one-hidden-layer feedforward networks. In contrast, our paper studies Transformer architectures under a different data model, and further incorporates data imbalance, providing a comprehensive analysis of how it influences the model's ability to decouple features, rather than being only a direct extension through feature magnitude changes.

# B  Preliminaries

**Data Model: Sparse Coding.**   For the necessity of theoretical proof, we adopt the sparse coding model [22, 23, 7, 36, 23, 26, 41, 20, 2] as a conceptual modeling of real-world data. Specifically, for a paired data $(X_n, X_{n'})$, the data structure is

$$
\begin{aligned}
X_n &= \left[ \mathbf{M}z_n^{(1)} + \xi_n^{(1)}, \ \mathbf{M}z_n^{(2)} + \xi_n^{(2)}, \ \ldots, \ \mathbf{M}z_n^{(L)} + \xi_n^{(L)} \right] \\
X_{n'} &= \left[ \mathbf{M}z_{n'}^{(1)} + \xi_{n'}^{(1)}, \ \mathbf{M}z_{n'}^{(2)} + \xi_{n'}^{(2)}, \ \ldots, \ \mathbf{M}z_{n'}^{(L)} + \xi_{n'}^{(L)} \right]
\end{aligned}
\tag{10}
$$

Here, each $z_n^{(i)} \in \mathbb{R}^d$ represents the latent signal at the $\ell$-th token, $\xi_n^{(i)}$ denotes the additive noise, and $\mathbf{M} = [\mathbf{M}_1, \ldots, \mathbf{M}_d] \in \mathbb{R}^{d_1 \times d}$ is the dictionary matrix. For each index, $z_{n,j}^{(i)} = 0$ indicates that the corresponding feature is absent in the token, while $z_{n,j}^{(i)} \neq 0$ indicates that the feature is present.

For a positive pair, we assume they share the same group of features when counting over all tokens in the sample, whereas negative samples are independent. That is to say, $\sum_{\ell=1}^{L} z_n^{(\ell)}$ and $\sum_{\ell=1}^{L} z_{n'}^{(\ell)}$ share the same support in a positive pair, while $z_n^{(\ell)}$ and $z_{n'}^{(\ell)}$ remain independent in a negative pair.

We first recall a useful concentration property. Whenever the Frobenius norm of the weight matrix satisfies:

$$
\| w^{(t)} \|_F^2 = \sum_{i \in [m]} \| w_i^{(t)} \|_2^2 \leq \text{poly}(d),
\tag{11}
$$

the following estimate can be obtained by applying Bernstein concentration inequalities.

**Fact B.1** (Approximation of population gradients by empirical gradients)**.** *Suppose that $\| w^{(t)} \|_F^2 \leq \text{poly}(d)$. Then there exists some $K = \text{poly}(d)$ such that, with high probability, the difference between*

*the empirical gradients and the population gradients is bounded for every iteration $t$:*

$$\left\|\nabla_{w_i}\widehat{\mathrm{Obj}}(f_t) - \nabla_{w_i}\mathrm{Obj}(f_t)\right\|_2 \leq \frac{\|w_i^{(t)}\|_2}{\mathrm{poly}(d_1)}, \quad \forall i \in [m]. \tag{12}$$

To facilitate the calculation of the gradient of the loss function $\mathcal{L}(f_t, X_n, \mathcal{B}^\ell)$ with respect to the weights $\{w_i^{(t)}\}_{i\in[m]}$, we introduce the following notation. We denote by $\ell'_{p,t}(X_n, \mathcal{B})$ the positive logit, and by $\ell'_{s,t}(X_n, \mathcal{B})$ the negative logits:

$$\ell'_{p,t}(X_n, \mathcal{B}) := \frac{\exp\left(\mathrm{Sim}_{f_t}(X_n, X_{n'})/\tau\right)}{\sum_{x\in\mathcal{B}}\exp\left(\mathrm{Sim}_{f_t}(X_n, x)/\tau\right)} \tag{13}$$

$$\ell'_{s,t}(X_n, \mathcal{B}) := \frac{\exp\left(\mathrm{Sim}_{f_t}(X_n, X_{n,s})/\tau\right)}{\sum_{x\in\mathcal{B}}\exp\left(\mathrm{Sim}_{f_t}(X_n, x)/\tau\right)} \tag{14}$$

The empirical gradient of $L(f_t)$ with respect to the weight $w_i^{(t)}$ at iteration $t$ is given by (note that the similarity measure $\mathrm{Sim}_{f_t}$ makes use of the StopGrad operation):

$$\nabla_{w_i}L(f_t) = \mathbb{E}\left[(\ell'_{p,t} - 1)h_i(X_{n'})\sum_{r=1}^{L}\mathbf{1}_{|\langle w_i, z_X^{(r)}\rangle|\geq b_i}z_X^{(r)} + \sum_{X_{n,s}\in\mathfrak{N}}\ell'_{s,t}h_i(X_{n,s})\sum_{r=1}^{L}\mathbf{1}_{|\langle w_i, z_X^{(r)}\rangle|\geq b_i}z_X^{(r)}\right] \tag{15}$$

## C Lemmas

**Definition C.1** (Characterization of Neurons). *We choose constants*

$$c_1 = \left(\frac{\epsilon_{\max}}{\epsilon_{\min}}\right)^2 \cdot 2(1 + \gamma c_0), \quad c_2 = \left(\frac{\epsilon_{\min}}{\epsilon_{\max}}\right)^2 \cdot 2(1 - \gamma c_0), \quad \gamma c_0 \in (0, 0.001)$$

*We define the following sets of neurons, which will be useful for analyzing the stochastic gradient descent trajectory in later sections:*

*For each $j \in [d]$, we define the set of* ordinary neurons $\mathcal{M}_j \subseteq [m]$ *as:*

$$\mathcal{M}_j := \left\{ i \in [m] : \langle w_i^{(0)}, \mathbf{M}_j \rangle^2 \geq \frac{c_2 \log d}{d} \|\mathbf{M}\mathbf{M}^\top w_i^{(0)}\|_2^2 \right\}, \quad \forall j \in [d] \tag{16}$$

*For each $j \in [d]$, we define the set of* lucky neurons $\mathcal{M}_j^\star \subseteq [m]$ *as:*

$$\mathcal{M}_j^\star := \left\{ \begin{matrix} i \in [m] : \langle w_i^{(0)}, \mathbf{M}_j \rangle^2 \geq \frac{c_1 \log d}{d} \|\mathbf{M}\mathbf{M}^\top w_i^{(0)}\|_2^2, \\ \langle w_i^{(0)}, \mathbf{M}_{j'} \rangle^2 \leq \frac{c_2 \log d}{d} \|\mathbf{M}\mathbf{M}^\top w_i^{(0)}\|_2^2, \quad \forall j' \in [d], j' \neq j \end{matrix} \right\}, \quad \forall j \in [d] \tag{17}$$

**Lemma C.1.** *At initialization ($t = 0$), the following properties hold:*

*(a) With high probability, for every $i \in [m]$,*

$$\|w_i^{(0)}\|_2^2 \in \left[ \sigma_0^2 d_1 \left( 1 - \widetilde{O}\left(\tfrac{1}{\sqrt{d_1}}\right) \right), \ \sigma_0^2 d_1 \left( 1 + \widetilde{O}\left(\tfrac{1}{\sqrt{d_1}}\right) \right) \right]. \tag{18}$$

*(b) With high probability, for every $i \in [m]$,*

$$\|\mathbf{M}\mathbf{M}^\top w_i^{(0)}\|_2^2 \in \left[ \sigma_0^2 d \left( 1 - \widetilde{O}\left(\tfrac{1}{\sqrt{d}}\right) \right), \ \sigma_0^2 d \left( 1 + \widetilde{O}\left(\tfrac{1}{\sqrt{d}}\right) \right) \right]. \tag{19}$$

*(c) Let $m = d^{C_m}$ be the number of neurons and we note $\omega_1 = C_m - \left(\frac{\epsilon_{\max}}{\epsilon_{\min}}\right)^2 (1 + \gamma c_0)$, $\omega_2 = C_m - \left(\frac{\epsilon_{\min}}{\epsilon_{\max}}\right)^2 (1 - \gamma c_0)$. With probability at least $1 - o\left(\frac{1}{d^4}\right)$, for each $j \in [d]$,*

$$|\mathcal{M}_j^\star| \geq \Omega(d^{\omega_1}) =: \Xi_1, \qquad |\mathcal{M}_j| \leq O(d^{\omega_2}) =: \Xi_2. \tag{20}$$

*(d) For each $i \in [m]$, define*

$$\Lambda_i := \left\{ j \in [d] : |\langle w_i^{(0)}, \mathbf{M}_j \rangle| \leq \tfrac{\sigma_0}{d} \right\} \subseteq [d]. \tag{21}$$

*Then*

$$|\Lambda_i| = O\left(\tfrac{d}{\text{polylog}(d)}\right). \tag{22}$$

*(e) For any $j' \neq j$, we have*

$$|\mathcal{M}_{j'} \cap \mathcal{M}_j| \leq O(\log d), \tag{23}$$

*with probability at least $1 - o(1/d^4)$.*

*(f) For each $i \in [m]$, there are at most $O(1)$ indices $j \in [d]$ such that $i \in \mathcal{M}_j$, and at most $O(2^{-\sqrt{\log d}} d)$ indices $j \in [d]$ such that*

$$|\langle w_i^{(0)}, \mathbf{M}_j \rangle| \geq \Omega(\sigma_0 \log^{1/4} d). \tag{24}$$

**Lemma C.2** (Pre-activation size I). *Let $z_X^{(r)} = \frac{1}{L}\left(\mathbf{M}\tilde{z}_n^{(r)} + \tilde{\xi}_n^{(r)}\right) \sim \mathcal{D}_{z_X}$, $w_i \in \mathbb{R}^{d_1}$. Define $z_X^{(r)\backslash j} = \frac{1}{L}\left(\sum_{j' \neq j, \, j' \in [d]} \mathbf{M}_{j'} \tilde{z}_{n,j'}^{(r)} + \tilde{\xi}_n^{(r)}\right)$. Then the following results hold:*

*(a) Naive Chebyshev bound: For any $\lambda > 0$,*

$$\Pr_{\tilde{z}_n^{(r)\backslash j}, \tilde{\xi}_n^{(r)}} \left( \left( \langle w_i, z_X^{(r)\backslash j} \rangle + \tfrac{1}{L}\langle w_i, M_j \rangle \tilde{z}_{n,j}^{(r)} \right)^2 > \tfrac{\lambda \|w_i\|_2^2 \sqrt{\log d}}{d} \right) \leq O\left(\tfrac{1}{\lambda}\right). \tag{25}$$

The same tail bound applies to $\langle w_i, z_X^{(r)} \rangle$, $\langle w_i, \frac{z_Y^{(s)} - z_X^{(r)}}{2} \rangle$, and $\langle w_i, \tilde{\xi}_n^{(r)} \rangle$.

*(b) High probability bound for sparse signal:*

$$\Pr\left( \langle w_i, \mathbf{M} \tilde{z}_n^{(r)} \rangle^2 > \|w_i\|_2^2 \cdot \max_{j \in [d]} \|\mathbf{M}_j\|_\infty^2 \log^4 d \right) \lesssim e^{-\Omega(\log^2 d)}. \tag{26}$$

*(c) High probability bound for dense signal: Let* $Z = \langle w_i, \tilde{\xi}_n^{(r)} \rangle$. *Then*

$$\Pr\left( Z^2 \geq \frac{\|w_i\|_2^2 \log^4 d}{d} \right) \lesssim e^{-\Omega(\log^2 d)}. \tag{27}$$

**Lemma C.3** (Pre-activation size II). *Suppose the following conditions hold:*

$$\langle w_i^{(t)}, \mathbf{M}_j \rangle^2 \geq \Omega\big((b_i^{(t)})^2\big) \quad \text{for at most } O(1) \text{ indices } j \in [d], \tag{28}$$

$$\langle w_i^{(t)}, \mathbf{M}_j \rangle^2 \geq \Omega\left( \frac{(b_i^{(t)})^2}{\sqrt{\log d}} \right) \quad \text{for at most } O\big(e^{-\Omega(\sqrt{\log d})} d\big) \text{ indices } j \in [d], \tag{29}$$

$$\|w_i^{(t)}\|_2^2 \leq O\left( \frac{d(b_i^{(t)})^2}{\log d} \right). \tag{30}$$

*Then, for any* $\lambda \geq 0.0001$,

$$\Pr\left( |\langle w_i^{(t)}, z_X^{(r)} \rangle| \geq \lambda b_i^{(t)} \right) \lesssim e^{-\Omega(\log^{1/4} d)}, \tag{31}$$

*and*

$$\Pr\left( \left| \left\langle w_i^{(t)}, \frac{z_X^{(r)} + z_X^{(s)}}{2} \right\rangle \right| \geq \lambda b_i^{(t)} \right) \lesssim e^{-\Omega(\log^{1/4} d)}. \tag{32}$$

**Lemma C.4** (Pre-activation size III). *Let* $i \in [m]$. *Suppose there exists a set* $\mathcal{N}_i \subseteq [d]$ *with* $|\mathcal{N}_i| = O(1)$ *such that*

$$\langle w_i^{(t)}, \mathbf{M}_j \rangle^2 \leq O\left( \frac{(b_i^{(t)})^2}{\text{polylog}(d)} \right), \quad \forall j \notin \mathcal{N}_i, \tag{33}$$

*and*

$$\|w_i^{(t)}\|_2^2 \leq O\left( \frac{d(b_i^{(t)})^2}{\text{polylog}(d)} \right). \tag{34}$$

*Then, for any* $\lambda \in [0.01, 0.99]$,

$$\Pr\left[ \left| \sum_{j \notin \mathcal{N}_i} \langle w_i^{(t)}, \mathbf{M}_j \rangle \tilde{z}_{n,j}^{(r)} + \langle w_i, \tilde{\xi}_n^{(r)} \rangle \right| \geq \lambda b_i^{(t)} \right] \lesssim e^{-\Omega(\log^2 d)}. \tag{35}$$

# D  Theorem 3.1

**Lemma D.1** (Positive gradient, stage I). *Let $h_{i,t}(\cdot)$ denote the $i$-th neuron at iteration $t \leq T_1$ (so that $b_i^{(t)} = 0$). Then the following hold:*

*(a) For each $j \in [d]$,*

$$\mathbb{E}[h_{i,t}(X_{n'}) \langle \nabla_{w_i} h_{i,t}(X_n), \mathbf{M}_j \rangle] = \frac{1}{L^2} \langle w_i^{(t)}, \mathbf{M}_j \rangle \, \mathbb{E}[\hat{z}_{n',j''} \hat{z}_{n,j}] \tag{36}$$

*(b) For each $j \in [d_1] \setminus [d]$,*

$$\mathbb{E}\big[h_{i,t}(X_{n'}) \langle \nabla_{w_i} h_{i,t}(X_n), \mathbf{M}_j^{\perp} \rangle\big] = 0 \tag{37}$$

**Lemma D.2** (Logits near initialization). *Let $w_i \in \mathbb{R}^{d_1}$ for each $i \in [m]$. Suppose*

$$\sum_{i \in [m]} \left\| w_i^{(t)} \right\|_2^2 \leq o\big(\tfrac{\tau}{d}\big). \tag{38}$$

*Then, with high probability over the randomness of $X_n$, $X_{n'}$, and $\mathfrak{N}$, it holds that*

$$\left| \ell'_{p,t}(X_n, \mathfrak{B}) - \tfrac{1}{|\mathfrak{B}|} \right| \cdot \left| \ell'_{s,t}(X_n, \mathfrak{B}) - \tfrac{1}{|\mathfrak{B}|} \right| \leq \widetilde{\mathcal{O}} \left( \frac{\sum_{i \in [m]} \left\| w_i^{(t)} \right\|_2^2}{\tau |\mathfrak{B}|} \right) \tag{39}$$

Recall that

$$T_1 = \Theta \left( \frac{d \log d}{\eta \log \log d} \right) \tag{40}$$

is defined as the iteration when

$$\left\| w_i^{(t)} \right\|_2^2 \geq (1 + \epsilon_{\min} C_z \log d)^2 \left\| w_i^{(0)} \right\|_2^2, \quad \forall i \in [m], \tag{41}$$

and such a $T_1$ is indeed of order $\Theta \left( \frac{d \log d}{\eta \log \log d} \right)$.

The gradient descent update for the projection of $w_i^{(t)}$ onto $\mathbf{M}_j$ can be written as

$$
\begin{aligned}
\langle w_i^{(t+1)}, \mathbf{M}_j \rangle &= \langle w_i^{(t)}, \mathbf{M}_j \rangle - \eta \langle \nabla_{w_i} \mathbf{Obj}(f_t), \mathbf{M}_j \rangle \;\; \pm \frac{\|w_i^{(t)}\|_2}{\text{poly}(d_1)} \\
&= (1 - \eta \lambda) \langle w_i^{(t)}, \mathbf{M}_j \rangle + \eta \, \mathbb{E}_{X_n, X_{n'}} \big[ (1 - \ell'_{p,t}(X_n, \mathfrak{B})) \cdot h_{i,t}(X_{n'}) \langle \nabla_{w_i} h_i(X_n), \mathbf{M}_j \rangle \big] \\
&\quad - \eta \sum_{X_{n,s} \in \mathfrak{N}} \mathbb{E}\big[ \ell'_{s,t}(X_n, \mathfrak{B}) \, h_{i,t}(X_{n,s}) \langle \nabla_{w_i} h(X_n), \mathbf{M}_j \rangle \big] \;\; \pm \frac{\|w_i^{(t)}\|_2}{\text{poly}(d_1)}
\end{aligned}
\tag{42}
$$

For the positive term: we can use Lemma D.1 and Lemma D.2 to obtain that:

$$\mathbb{E}\big[(1 - \ell'_{p,t}(X_n, \mathfrak{B})) \cdot h_{i,t}(X_{n'}) \langle \nabla_{w_i} h_{i,t}(X_n), \mathbf{M}_j \rangle \big] = \frac{1}{L^2} \langle w_i^{(t)}, \mathbf{M}_j \rangle \, \mathbb{E}[\hat{z}_{n',j} \hat{z}_{n,j}] \tag{43}$$

For the negative term: Here, the bound needs to be verified because Lemma D.2.

$$
\begin{aligned}
\mathbb{E}\left[ \sum_{X \in \mathfrak{N}} \ell'_{s,t} \, h_{i,t}(X) \langle \nabla_{w_i} h(X_n), \mathbf{M}_j \rangle \right] &\overset{(1)}{=} \sum_{X \in \mathfrak{N}} \mathbb{E}\left[ \left( \ell'_{s,t} - \tfrac{1}{|\mathfrak{B}|} \right) h_{i,t}(X) \langle \nabla_{w_i} h(X_n), \mathbf{M}_j \rangle \right] \\
&\overset{(2)}{\leq} \sum_{X \in \mathfrak{N}} \mathbb{E}\left[ \left| \ell'_{s,t} - \tfrac{1}{|\mathfrak{B}|} \right| \cdot |h_{i,t}(X)| \cdot \left| \langle \nabla_{w_i} h(X_n), \mathbf{M}_j \rangle \right| \right] \\
&\overset{(3)}{\leq} \widetilde{\mathcal{O}}\left( \frac{\sum_{i \in [m]} \|w_i^{(t)}\|_2^2}{\tau d} \cdot \|w_i^{(t)}\|_2 \right)
\end{aligned}
\tag{44}
$$

Putting all the above calculations together, we have

$$\langle w_i^{(t+1)}, \mathbf{M}_j \rangle = \left(1 - \eta\lambda + \epsilon_j \frac{\eta C_z \log\log d}{d}\right) \langle w_i^{(t)}, \mathbf{M}_j \rangle$$
$$\pm \widetilde{O}\left(\frac{\eta \sum_{i\in[m]} \|w_i^{(t)}\|_2^2}{\tau d} \cdot \|w_i^{(t)}\|_2\right) \pm \widetilde{O}\left(\frac{\|w_i^{(t)}\|_2}{\mathrm{poly}(d_1)}\right) \tag{45}$$

Prior to the induction step, we establish, by a similar method, the stochastic gradient descent update of $w_i$ along the dense feature direction $\mathbf{M}_j^\perp$. Specifically, we obtain the following update equation:

$$\langle w_i^{(t+1)}, \mathbf{M}_j^\perp \rangle = \langle w_i^{(t)}, \mathbf{M}_j^\perp \rangle - \eta\langle \nabla_{w_i}\mathbf{Obj}(f_t), \mathbf{M}_j^\perp \rangle$$
$$= (1 - \eta\lambda)\langle w_i^{(t)}, \mathbf{M}_j^\perp \rangle + \eta\, \mathbb{E}\big[(1 - \ell'_{p,t})\, h_{i,t}(x_p^{++})\langle \nabla_{w_i} h_{i,t}(x_p^+), \mathbf{M}_j^\perp \rangle\big]$$
$$- \eta \sum_{x_{n,s}\in\mathfrak{N}} \mathbb{E}\big[\ell'_{s,t}\, h_{i,t}(x_{n,s})\langle \nabla_{w_i} h(x_p^+), \mathbf{M}_j^\perp \rangle\big] + \frac{\|w_i^{(t)}\|_2}{\mathrm{poly}(d_1)}$$
$$= (1 - \eta\lambda)\langle w_i^{(t)}, \mathbf{M}_j^\perp \rangle \pm \widetilde{O}\left(\frac{\eta \sum_{i\in[m]} \|w_i^{(t)}\|_2^2}{\tau d} \cdot \|w_i^{(t)}\|_2\right) \pm \widetilde{O}\left(\frac{\|w_i^{(t)}\|_2}{\mathrm{poly}(d_1)}\right) \tag{46}$$

*Proof of Theorem 3.1.* For $j \in [d]$ and $i \in [m]$, at iteration $T_1$ the following bounds hold:

(a) Lower bound:

$$|\langle w_i^{(T_1)}, \mathbf{M}_j \rangle| \geq |\langle w_i^{(0)}, \mathbf{M}_j \rangle| \left(1 - \eta\lambda + \epsilon_j \frac{\eta C_z \log\log d}{d}\right)^{T_1} - \widetilde{O}\left(\frac{\eta T_1 \|w_i^{(T_1)}\|_2}{d_1}\right) \tag{47}$$

(b) Upper bound:

$$|\langle w_i^{(T_1)}, \mathbf{M}_j \rangle| \leq |\langle w_i^{(0)}, \mathbf{M}_j \rangle| \left(1 + \epsilon_j \frac{\eta C_z \log\log d}{d} + \widetilde{O}\left(\frac{\eta}{d^2}\right)\right)^{T_1} + \widetilde{O}\left(\frac{\eta T_1 \|w_i^{(T_1)}\|_2}{d_1}\right) \tag{48}$$

(c) Orthogonal component:

$$|\langle w_i^{(T_1)}, \mathbf{M}_j^\perp \rangle| \leq |\langle w_i^{(0)}, \mathbf{M}_j^\perp \rangle| + O(T_1\eta) \cdot \max_{t\leq T_1} O\left(\frac{\|w_i^{(t)}\|_2}{d_1}\right) \tag{49}$$

$\square$

The proof follows by iterating the gradient descent update for $w_i$ along the signal direction $\mathbf{M}_j$ and its orthogonal complement, while controlling the error terms at each step. By substituting $T_1$ into the recurrence, the bounds in 3.1 follow directly.

# E    Theorem 3.2

In this part, we analyze how each feature $\mathcal{M}_j$ may be captured by certain subsets of neurons, a process that is influenced by the stochastic nature of initialization.

**Lemma E.1.** *For all iterations $t \in (T_1, T_2]$, the neurons $i \in [m]$ satisfy the following properties:*

*(a) For $j \in [d]$, if $i \in \mathcal{M}_j^\star$, then*

$$\left| \langle w_i^{(t)}, \mathbf{M}_j \rangle \right| \geq \sqrt{1 + \gamma c_0} \, b_i^{(t)} \tag{50}$$

*(b) For $j \in [d]$, if $i \notin \mathcal{M}_j$, then*

$$\left| \langle w_i^{(t)}, \mathbf{M}_j \rangle \right| \leq \sqrt{1 - \gamma c_0} \, b_i^{(t)} \tag{51}$$

*and furthermore,*

$$\left| \langle w_i^{(t)}, \mathbf{M}_j \rangle \right| \leq \widetilde{\mathcal{O}}\left( \frac{\|w_i^{(t)}\|_2}{\sqrt{d}} \right) \tag{52}$$

*(c) For each $i \in [m]$, there are at most $\mathcal{O}(2^{-\sqrt{\log d}}d)$ many $j \in [d]$ such that*

$$\langle w_i^{(t)}, \mathbf{M}_j \rangle^2 \geq \frac{(b_i^{(t)})^2}{\sqrt{\log d}} \tag{53}$$

*(d) For each $i \in [m]$, and for all $j \in [d_1] \setminus [d]$,*

$$\left| \langle w_i^{(t)}, \mathbf{M}_j^\perp \rangle \right| \leq \widetilde{\mathcal{O}}\left( \frac{\|w_i^{(t)}\|_2}{\sqrt{d_1}} \right) \tag{54}$$

*(e) For all $i \in [m]$,*

$$\|w_i^{(t)}\|_2^2 \leq \frac{d(b_i^{(t)})^2}{\log d} \tag{55}$$

**Definition E.1** (Notations). *For simpler presentation, we define the following notations: given $z_X = \frac{1}{L}(\mathbf{M}\tilde{z}_n + \tilde{\xi}_n) \sim \mathcal{D}_{z_X}$, $z_Y = \frac{1}{L}(\mathbf{M}\tilde{z}_{n'} + \tilde{\xi}_{n'}) \sim \mathcal{D}_{z_Y}$, we let (for each $j \in [d]$):*

$$z_X^{\backslash j} := \frac{1}{L}\left( \sum_{\substack{j' \neq j \\ j' \in [d]}} \mathbf{M}_{j'} \tilde{z}_{n,j'} + \tilde{\xi}_n \right), \quad z_Y^{\backslash j} := \frac{1}{L}\left( \sum_{\substack{j' \neq j \\ j' \in [d]}} \mathbf{M}_{j'} \tilde{z}_{n',j'} + \tilde{\xi}_{n'} \right) \tag{56}$$

$$S_{i,t}^{(r)\backslash j} := \langle w_i^{(t)}, z_X^{(r)\backslash j} \rangle, \quad S_{i,t}^{(s)\backslash j} := \langle w_i^{(t)}, z_Y^{(s)\backslash j} \rangle \tag{57}$$

$$S_{i,t}^{(r,s)\backslash j} := \frac{1}{2}\left( S_{i,t}^{(r)\backslash j} + S_{i,t}^{(s)\backslash j} \right), \quad \bar{S}_{i,t}^{(r,s)\backslash j} := \frac{1}{2}\left( S_{i,t}^{(s)\backslash j} - S_{i,t}^{(r)\backslash j} \right) \tag{58}$$

$$\alpha_{i,j}^{(t)} := \langle w_i^{(t)}, \mathbf{M}_j \rangle, \quad \bar{\alpha}_{i,j}^{(r,s)(t)} := \left\langle w_i^{(t)}, \frac{\tilde{z}_{n',j}^{(s)} - \tilde{z}_{n,j}^{(r)}}{\tilde{z}_{n,j}^{(r)} + \tilde{z}_{n',j}^{(s)}} \mathbf{M}_j \right\rangle \tag{59}$$

Whenever the neuron index $i \in [m]$ is clear from the context, we drop the subscript $i$ and the time index $t$ for notational simplicity.

**Lemma E.2** (Gradient for sparse features). *Suppose E.1 holds at iteration $t \geq 0$. For $j \in [d]$, we denote events*

$$
\begin{aligned}
A_1 &:= \left\{ S_{i,t}^{\setminus j} \geq b_i^{(t)} - \alpha_{i,j}^{(t)} C_{\tilde{z}} \right\}, \\
A_2 &:= \left\{ \bar{S}_{i,t}^{\setminus j} \geq b_i^{(t)} - \bar{\alpha}_{i,j}^{(t)} C_{\tilde{z}} \right\}, \\
A_3 &:= \left\{ \left| \bar{S}_{i,t}^{\setminus j} + \bar{\alpha}_{i,j}^{(t)} C_{\tilde{z}} \right| \geq \tfrac{1}{2} \left( \alpha_{i,j}^{(t)} C_{\tilde{z}} - b_i^{(t)} \right) \right\}, \\
A_4 &:= \left\{ S_{i,t}^{\setminus j} \geq \tfrac{1}{2} \left( \alpha_{i,j}^{(t)} C_{\tilde{z}} - b_i^{(t)} \right) \right\};
\end{aligned}
\tag{60}
$$

*and quantities $L_1, L_2, L_3, L_4$ as*

$$
L_1 := \sqrt{ \frac{ \mathbb{E}[|\bar{S}_{i,t}^{\setminus j}|^2 (\mathbf{1}_{A_1} + \mathbf{1}_{A_2})] }{ \mathbb{E}[\langle w_i^{(t)}, \tilde{\xi} \rangle^2] } }, \quad L_2 := \Pr(A_1), \quad L_3 := \sqrt{ \frac{ \mathbb{E}[|\bar{S}_{i,t}^{\setminus j}|^2 (\mathbf{1}_{A_3} + \mathbf{1}_{A_4})] }{ \mathbb{E}[\langle w_i^{(t)}, \tilde{\xi} \rangle^2] } }, \quad L_4 := \Pr(A_3)
\tag{61}
$$

*Then we have the following results:*

*(a) (all features) For all $i \in [m]$, if $\alpha_{i,j}^{(t)} \geq 0$, we have (when $\alpha_{i,j}^{(t)} \leq 0$ the opposite inequality holds)*

$$
\begin{aligned}
\mathbb{E}\left[ h_i(X_{n'}) \sum_{r=1}^{L} \mathbf{1}_{|\langle w_i, z_X^{(r)} \rangle| \geq b_i} \tilde{z}_{n,j}^{(r)} \right] &\leq \frac{1}{L} \alpha_{i,j}^{(t)} \cdot \mathbb{E}\left[ \sum_{s=1}^{L} \tilde{z}_{n',j}^{(s)} \sum_{r=1}^{L} \tilde{z}_{n,j}^{(r)} \mathbf{1}_{|\langle w_i^{(t)}, \frac{z_X + z_Y}{2} \rangle| \geq b_i + |\langle w_i^{(t)}, z_X - \frac{z_X + z_Y}{2} \rangle|} \right] \\
&\quad \pm \left( \alpha_{i,j}^{(t)} + O\left( \sqrt{ \mathbb{E}|\bar{\alpha}_{i,j}^{(t)}|^2 } \right) \right) \cdot \mathbb{E}\left[ \sum_{s=1}^{L} \sum_{r=1}^{L} \left| \frac{ \tilde{z}_{n,j}^{(r)} + \tilde{z}_{n',j}^{(s)} }{2} \right| |\tilde{z}_{n,j}^{(r)}| \right] \cdot O(L_1 + L_2)
\end{aligned}
\tag{62}
$$

*(b) (lucky features) If $\alpha_{i,j}^{(t)} > b_i^{(t)}$, we have*

$$
\begin{aligned}
\mathbb{E}\left[ h_i(X_{n'}) \sum_{r=1}^{L} \mathbf{1}_{|\langle w_i, z_X^{(r)} \rangle| \geq b_i} \tilde{z}_{n,j}^{(r)} \right] &\leq \frac{1}{L} \left( \alpha_{i,j}^{(t)} - b_i^{(t)} \right) \cdot \mathbb{E}\left[ \sum_{s=1}^{L} \tilde{z}_{n',j}^{(s)} \sum_{r=1}^{L} \tilde{z}_{n,j}^{(r)} \mathbf{1}_{|\langle w_i^{(t)}, \frac{z_X + z_Y}{2} \rangle| \geq b_i + |\langle w_i^{(t)}, z_X - \frac{z_X + z_Y}{2} \rangle|} \right] \\
&\quad \pm \left( \alpha_{i,j}^{(t)} + O\left( \sqrt{ \mathbb{E}|\bar{\alpha}_{i,j}^{(t)}|^2 } \right) \right) \cdot \mathbb{E}\left[ \sum_{s=1}^{L} \sum_{r=1}^{L} \left| \frac{ \tilde{z}_{n,j}^{(r)} + \tilde{z}_{n',j}^{(s)} }{2} \right| |\tilde{z}_{n,j}^{(r)}| \right] \cdot O(L_3 + L_4)
\end{aligned}
\tag{63}
$$

*If $\alpha_{i,j}^{(t)} < -b_i^{(t)}$, then the opposite inequality holds with $(\alpha_{i,j}^{(t)} - b_i^{(t)})$ replaced by $(\alpha_{i,j}^{(t)} + b_i^{(t)})$*

**Lemma E.3** (Gradient from dense signals). *Let $i \in [m]$ and $j \in [d]$. Suppose E.1 holds for the current iteration $t$. Then*

$$
\left| \mathbb{E}\left[ h_i(X_{n'}) \sum_{r=1}^{L} \mathbf{1}_{|\langle w_i^{(t)}, z_X^{(r)} \rangle| \geq b_i^{(t)}} \langle \tilde{\xi}_n^{(r)}, \mathbf{M}_j \rangle \right] \right| \leq \widetilde{O}\left( \frac{ \|w_i^{(t)}\|_2 }{ d^2 } \right) \cdot \Pr\left( h_{i,t}(X_{n'}) \neq 0 \right)
\tag{64}
$$

*For dense features $\mathbf{M}_j^{\perp}$, $j \in [d_1] \setminus [d]$, we have a similar result:*

$$
\left| \mathbb{E}\left[ h_i(X_{n'}) \sum_{r=1}^{L} \mathbf{1}_{|\langle w_i^{(t)}, z_X^{(r)} \rangle| \geq b_i^{(t)}} \langle \tilde{\xi}_n^{(r)}, \mathbf{M}_j^{\perp} \rangle \right] \right| \leq \widetilde{O}\left( \frac{ \|w_i^{(t)}\|_2 }{ d\sqrt{d_1} } \right) \cdot \Pr\left( h_{i,t}(X_{n'}) \neq 0 \right)
\tag{65}
$$

The second stage is defined as the iterations $t \geq T_1$ but $t \leq T_2$, where

$$
T_2 = \Theta\left( \frac{ d \log d }{ \epsilon_{\max} \eta \log \log d } \right)
\tag{66}
$$

is defined as the iteration when one of the neuron $i \in [m]$ satisfies

$$
\left\| w_i^{(T_2)} \right\|_2^2 \geq d \left\| w_i^{(T_1)} \right\|_2^2
\tag{67}
$$

**Now we separately discuss three cases:**

(a) When $i \in \mathcal{M}_j^\star$, if $\tilde{z}_{n',j}^{(s)}$ and $\tilde{z}_{n,j}^{(r)} \neq 0$, say $\frac{\tilde{z}_{n',j}^{(s)} + \tilde{z}_{n,j}^{(r)}}{2} = C_{\tilde{z}}^{(r,s)}$, we simply have

$$
\mathbb{E}\left[\sum_{s=1}^{L} \tilde{z}_{n',j}^{(s)} \sum_{r=1}^{L} \tilde{z}_{n,j}^{(r)} \mathbf{1}_{\left|\langle w_i, \frac{z_X^{(r)} + z_Y^{(s)}}{2}\rangle\right| \geq b_i + \left|\langle w_i, z_X^{(r)} - \frac{z_X^{(r)} + z_Y^{(s)}}{2}\rangle\right|}\right]
$$

$$
= \mathbb{E}\left[\sum_{s=1}^{L} \tilde{z}_{n',j}^{(s)} \sum_{r=1}^{L} \tilde{z}_{n,j}^{(r)}\right] \cdot \Pr\left(\left|\langle w_i^{(t)}, \frac{z_X^{(r)} + z_Y^{(s)}}{2}\rangle\right| \geq b_i + \left|\langle w_i^{(t)}, z_X^{(r)} - \frac{z_X^{(r)} + z_Y^{(s)}}{2}\rangle\right|\right) \quad (68)
$$

$$
= \epsilon_j \frac{L^2 C_z \log\log d}{d}\left(1 - \frac{1}{\text{polylog}(d)}\right).
$$

For $\mathbf{M}_j$ such that $i \in \mathcal{M}_j^\star$, at iteration $t + 1$:

$$
\langle w_i^{(t+1)}, \mathbf{M}_j\rangle = \langle w_i^{(t)}, \mathbf{M}_j\rangle - \eta\langle\nabla_{w_i}\mathbf{Obj}(f_t), \mathbf{M}_j\rangle \pm \frac{\eta\|w_i^{(t)}\|_2}{\text{poly}(d_1)}
$$

$$
= \langle w_i^{(t)}, \mathbf{M}_j\rangle(1 - \eta\lambda) \pm \frac{\eta\|w_i^{(t)}\|_2}{\text{poly}(d_1)}
$$

$$
+ \eta\mathbb{E}\left[(1 - \ell_{p,t}^t)h_{i,t}(X_{n'})\sum_{r=1}^{L}\mathbf{1}_{|\langle w_i, z_X^{(r)}\rangle| \geq b_i}\langle z_X^{(r)}, \mathbf{M}_j\rangle\right]
$$

$$
- \eta\mathbb{E}\left[\sum_{X \in \mathfrak{N}} \ell_{s,t}^t h_{i,t}(X)\sum_{r=1}^{L}\mathbf{1}_{|\langle w_i, z_X^{(r)}\rangle| \geq b_i}\langle z_X^{(r)}, \mathbf{M}_j\rangle\right]
$$

$$
= \langle w_i^{(t)}, \mathbf{M}_j\rangle(1 - \eta\lambda) \pm \frac{\eta\|w_i^{(t)}\|_2}{\text{poly}(d_1)}
$$

$$
+ \eta\frac{1}{L}\mathbb{E}\left[(1 - \ell_{p,t}^t)h_{i,t}(X_{n'})\sum_{r=1}^{L}\mathbf{1}_{|\langle w_i, z_X^{(r)}\rangle| \geq b_i}\left(\tilde{z}_{n,j}^{(r)} + \langle\tilde{\xi}_n^{(r)}, \mathbf{M}_j\rangle\right)\right]
$$

$$
- \eta\frac{1}{L}\mathbb{E}\left[\sum_{X \in \mathfrak{N}} \ell_{s,t}^t h_{i,t}(X)\sum_{r=1}^{L}\mathbf{1}_{|\langle w_i, z_X^{(r)}\rangle| \geq b_i}\left(\tilde{z}_{n,j}^{(r)} + \langle\tilde{\xi}_n^{(r)}, \mathbf{M}_j\rangle\right)\right]
$$

$$
\geq \left(\langle w_i^{(t)}, \mathbf{M}_j\rangle - \text{sign}(\langle w_i^{(t)}, \mathbf{M}_j\rangle)\cdot b_i^{(t)}\right)\cdot\left(1 + \epsilon_j\frac{\eta C_z \log\log d}{d}\left(1 - \frac{\eta}{\text{polylog}(d)}\right)\right)
$$
$$(69)$$

**Next we compare this growth to the growth of bias $b_i^{(t+1)}$. Since we raise our bias by**

$$
b_i^{(t+1)} = \max\left\{b_i^{(t)}\left(1 + \frac{\eta}{d}\right),\ b_i^{(t)}\frac{\|w_i^{(t+1)}\|_2}{\|w_i^{(t)}\|_2}\right\} \quad (70)
$$

We have to prove

$$
\frac{|\langle w_i^{(t+1)}, \mathbf{M}_j\rangle|}{|\langle w_i^{(t)}, \mathbf{M}_j\rangle|} \geq \frac{\|w_i^{(t+1)}\|_2}{\|w_i^{(t)}\|_2}, \qquad i \in \mathcal{M}_j^\star \quad (71)
$$

We argue as follows: from previous calculations we have

$$
\sum_{j' \in [d],\, j' \neq j} \langle w_i^{(t+1)}, \mathbf{M}_{j'}\rangle^2 + \sum_{j' \in [d_1]\setminus[d]} \langle w_i^{(t+1)}, \mathbf{M}_{j'}^\perp\rangle^2
$$

$$
\leq \sum_{j' \in [d],\, j' \neq j} \langle w_i^{(t)}, \mathbf{M}_{j'}\rangle^2\left(1 + \epsilon_{j'}\frac{O(\eta)}{d\text{polylog}(d)}\right)^2 \quad (72)
$$

$$
+ \sum_{j' \in [d_1]\setminus[d]} \langle w_i^{(t)}, \mathbf{M}_{j'}^\perp\rangle^2 + \tilde{\mathcal{O}}\left(\frac{\eta}{d}\right)e^{-\Omega(\log^{1/4} d)}\|w_i^{(t)}\|_2^2
$$

Therefore by adding $\langle w_i^{(t+1)}, \mathbf{M}_j \rangle^2$ to the LHS we have

$$\|w_i^{(t+1)}\|_2^2 \leq \|w_i^{(t)}\|_2^2 \left(1 + \epsilon_{\max} \frac{O(\eta)}{d \cdot \text{polylog}(d)}\right)^2 + \left(\frac{|\langle w_i^{(t+1)}, \mathbf{M}_j \rangle|}{|\langle w_i^{(t)}, \mathbf{M}_j \rangle|} - \frac{O(\eta)}{d \cdot \text{polylog}(d)}\right) |\langle w_i^{(t)}, \mathbf{M}_j \rangle|^2 \tag{73}$$

which implies

$$\frac{\|w_i^{(t+1)}\|_2^2}{\|w_i^{(t)}\|_2^2} \leq \left(1 + \epsilon_{\max} \frac{O(\eta)}{d \cdot \text{polylog}(d)}\right)^2 + \left(\frac{|\langle w_i^{(t+1)}, \mathbf{M}_j \rangle|}{|\langle w_i^{(t)}, \mathbf{M}_j \rangle|}\right) \frac{|\langle w_i^{(t)}, \mathbf{M}_j \rangle|^2}{\|w_i^{(t)}\|_2^2} \tag{74}$$

Therefore,

$$\frac{|\langle w_i^{(t+1)}, \mathbf{M}_j \rangle|}{|\langle w_i^{(t)}, \mathbf{M}_j \rangle|} \geq \frac{\|w_i^{(t+1)}\|_2}{\|w_i^{(t)}\|_2} \tag{75}$$

as desired.

(b) When $i \notin \mathcal{M}_j$, we can similarly obtain that

$$\mathbb{E}\left[\sum_{s=1}^{L} \tilde{z}_{n',j}^{(s)} \sum_{r=1}^{L} \tilde{z}_{n,j}^{(r)} \mathbf{1}_{\left|\langle w_i, \frac{z_X^{(r)} + z_Y^{(s)}}{2}\rangle\right| \geq b_i + \left|\langle w_i, z_X^{(r)} - \frac{z_X^{(r)} + z_Y^{(s)}}{2}\rangle\right|}\right] \leq \epsilon_j \frac{L^2 C_z \log\log d}{d} \left(\frac{1}{\text{polylog}(d)}\right)$$

$$= O\left(\epsilon_j \frac{L^2}{d \cdot \text{polylog}(d)}\right) \tag{76}$$

And similarly we can compute the gradient descent dynamics as follows: For $j \in [d]$ such that $|\langle w_i^{(t)}, \mathbf{M}_j \rangle| \geq \frac{\|w_i^{(t)}\|_2 d}{\sqrt{d_1}}$, we have (assume here $\langle w_i^{(t)}, \mathbf{M}_j \rangle > 0$, the opposite is similar)

$$\langle w_i^{(t+1)}, \mathbf{M}_j \rangle = \langle w_i^{(t)}, \mathbf{M}_j \rangle - \eta \langle \nabla_{w_i} \mathbf{Obj}(f_t), \mathbf{M}_j \rangle + \frac{\eta \|w_i^{(t)}\|_2}{\text{poly}(d_1)}$$

$$\leq \langle w_i^{(t)}, \mathbf{M}_j \rangle \left(1 - \eta\lambda + \epsilon_j \frac{O(\eta)}{d \cdot \text{polylog}(d)}\right)$$

$$\pm O\left(\frac{\eta \sum_{i' \in [m]} \|w_{i'}^{(t)}\|_2^2 \|w_i^{(t)}\|_2}{d\tau}\right) \pm \widetilde{O}\left(\frac{\eta \|w_i^{(t)}\|_2}{d^2}\right) \tag{77}$$

$$\leq \langle w_i^{(t)}, \mathbf{M}_j \rangle \left(1 + \epsilon_j \frac{O(\eta)}{d \cdot \text{polylog}(d)}\right) + \widetilde{O}\left(\frac{\eta \|w_i^{(t)}\|_2}{d^2}\right)$$

It is also worth noting that similar calculations also lead to a lower bound:

$$|\langle w_i^{(t+1)}, \mathbf{M}_j \rangle| \geq |\langle w_i^{(t)}, \mathbf{M}_j \rangle|(1 - \eta\lambda) - \widetilde{O}\left(\eta \frac{\|w_i^{(t)}\|_2}{d^2}\right) \tag{78}$$

(c) Next we consider the learning dynamics for the dense features.

We can use E.3 to calculate its dynamics by

$$
\begin{aligned}
\langle w_i^{(t+1)}, \mathbf{M}_j^\perp \rangle &= \langle w_i^{(t)}, \mathbf{M}_j^\perp \rangle (1 - \eta\lambda) \pm \frac{\eta \|w_i^{(t)}\|_2}{\mathrm{poly}(d_1)} \\
&\quad + \eta \mathbb{E}\left[ (1 - \ell'_{p,t}) h_{i,t}(X_{n'}) \sum_{r=1}^{L} \mathbf{1}_{|\langle w_i, z_X^{(r)} \rangle| \geq b_i} \langle z_X^{(r)}, \mathbf{M}_j^\perp \rangle \right] \\
&\quad - \eta \sum_{X \in \mathfrak{N}} \mathbb{E}\left[ \ell'_{s,t} h_{i,t}(X) \sum_{r=1}^{L} \mathbf{1}_{|\langle w_i, z_X^{(r)} \rangle| \geq b_i} \langle z_X^{(r)}, \mathbf{M}_j^\perp \rangle \right] \\
&= \langle w_i^{(t)}, \mathbf{M}_j^\perp \rangle (1 - \eta\lambda) \pm \frac{\eta \|w_i^{(t)}\|_2}{\mathrm{poly}(d_1)} \\
&\quad + \eta \mathbb{E}\left[ (1 - \ell'_{p,t}) h_{i,t}(X_{n'}) \sum_{r=1}^{L} \mathbf{1}_{|\langle w_i, z_X^{(r)} \rangle| \geq b_i} \langle \tilde{\xi}_n^{(r)}, \mathbf{M}_j^\perp \rangle \right] \\
&\quad - \eta \sum_{X \in \mathfrak{N}} \mathbb{E}\left[ \ell'_{s,t} h_{i,t}(X) \sum_{r=1}^{L} \mathbf{1}_{|\langle w_i, z_X^{(r)} \rangle| \geq b_i} \langle \tilde{\xi}_n^{(r)}, \mathbf{M}_j^\perp \rangle \right] \\
&= \langle w_i^{(t)}, \mathbf{M}_j^\perp \rangle (1 - \eta\lambda) + \tilde{\mathcal{O}}\left( \frac{\eta \|w_i^{(t)}\|_2}{d\sqrt{d_1}} \right) \cdot \Pr(h_{i,t}(X_{n'}) \neq 0) \\
&\leq \langle w_i^{(t)}, \mathbf{M}_j^\perp \rangle + O\left( \frac{\eta \|w_i^{(t)}\|_2}{d\sqrt{d_1}} e^{-\Omega(\log^{1/4} d)} \right)
\end{aligned}
\tag{79}
$$

In the proof above, we have depended on the crucial assumption that $T_2 := \min\left\{ t \in \mathbb{N} : \exists i \in [m] \text{ s.t. } \|w_i^{(t)}\|_2^2 \geq d\|w_i^{(T_1)}\|_2^2 \right\}$ is of order $\Theta\left( \frac{d\log d}{\epsilon_{\max} \eta \log\log d} \right)$. Now we verify it as follows. If $i \in \mathcal{M}_j^*$ for some $j \in [d]$ (which also means $j' \notin \mathcal{N}_i$ for $j' \neq j$), we have

$$
\begin{aligned}
|\langle w_i^{(t)}, \mathbf{M}_j \rangle| &\geq |\langle w_i^{(T_1)}, \mathbf{M}_j \rangle| \left( 1 + \Omega\left( \epsilon_j \frac{\eta \log\log d}{d} \right) \right)^{t-T_1} \\
&\geq d\sqrt{\frac{2\log d}{d}} \|w_i^{(T_1)}\|_2 \quad \text{for some } t = O\left( \frac{d\log d}{\epsilon_j \eta \log\log d} \right)
\end{aligned}
\tag{80}
$$

Thus for some $t = O\left( \frac{d\log d}{\epsilon_j \eta \log\log d} \right)$, we have $|\langle w_i^{(t)}, \mathbf{M}_j \rangle|^2 \geq d\|w_i^{(T_1)}\|_2^2$, which proves that $T_2 \leq O\left( \frac{d\log d}{\epsilon_{\max} \eta \log\log d} \right)$.

Conversely, we also have for all $t \leq O\left( \frac{d\log d}{\epsilon_j \eta \log\log d} \right)$

$$
\begin{aligned}
&\sum_{j' \in [d]:j' \neq j} \langle w_i^{(t)}, \mathbf{M}_{j'} \rangle^2 + \sum_{j' \in [d_1]\setminus[d]} \langle w_i^{(t)}, \mathbf{M}_{j'}^\perp \rangle^2 \\
&\leq \|w_i^{(T_1)}\|_2^2 \left( 1 + \epsilon_{j'} \frac{O(\eta)}{d\,\mathrm{polylog}(d)} \right)^{t-T_1} + \max_{t' \leq t} O\left( \frac{\eta(t - T_1)}{d} \right) e^{-\Omega(\log^{1/4} d)} \|w_i^{(t')}\|_2^2 \\
&\leq o\left( d\|w_i^{(T_1)}\|_2^2 \right)
\end{aligned}
\tag{81}
$$

Except for the principal direction $\mathbf{M}_j$ (i.e., the alignment direction of neuron i ), the total growth of squared weights along all other directions remains far below the target scale $d \cdot \left\| w_i^{(T_1)} \right\|_2^2$. And also

$$
\begin{aligned}
|\langle w_i^{(t)}, \mathbf{M}_j \rangle| &\leq |\langle w_i^{(T_1)}, \mathbf{M}_j \rangle| \left( 1 + \epsilon_j \frac{C_z \eta \log\log d}{d} \left( 1 - \frac{1}{\mathrm{polylog}(d)} \right) \right)^{t-T_1} \\
&\leq O\left( \sqrt{\frac{\log d}{d}} \|w_i^{(T_1)}\|_2 \right) \left( 1 + \epsilon_j \frac{C_z \eta \log\log d}{d} \left( 1 - \frac{1}{\mathrm{polylog}(d)} \right) \right)^{t-T_1}
\end{aligned}
\tag{82}
$$

Therefore we at least need $\frac{d\log\left(\Omega\left(\sqrt{d}\sqrt{\frac{d}{\log d}}\right)\right)}{\epsilon_{\max}\eta C_z \log\log d}(1-o(1))$ iteration to let any neuron $i \in [m]$ reach $\|w_i^{(t)}\|_2^2 \ge d\|w_i^{(T_1)}\|_2$, which proves that $T_2 = \Theta\left(\frac{d\log d}{\epsilon_{\max}\eta \log\log d}\right)$

*Proof of Theorem 3.2.* **When all** $\|w_i^{(t)}\|_2 \le 2\|w_i^{(T_1)}\|_2$: The iteration complexity for a neuron $i \in [m]$ to reach $\|w_i^{(t)}\|_2 \ge 2\|w_i^{(T_1)}\|_2$ is no smaller than

$$T'_{i,1} := \max\left\{\Omega\left(\frac{d\log d}{\eta \log\log d}\right), T_2\right\}. \tag{83}$$

**For $j \in [d_1] \setminus [d]$ we have**

$$
\begin{aligned}
\sum_{j\in[d_1]\setminus[d]} \langle w_i^{(T'_{i,1})}, \mathbf{M}_j^\perp\rangle^2 &\le \sum_{j\in[d_1]\setminus[d]} \langle w_i^{(T_1)}, \mathbf{M}_j^\perp\rangle^2 + O\left(\frac{\eta(T'_{i,1}-T_1)}{d}\right) e^{-\Omega(\log^{1/4} d)} \max_{t'\in[T_1,T'_{i,1}]} \|w_i^{(t')}\|_2^2 \\
&\le (1+o(1))\left\|\mathbf{M}^\perp(\mathbf{M}^\perp)^\top w_i^{(T_1)}\right\|_2^2
\end{aligned}
\tag{84}
$$

If $i \in \mathcal{M}_j^*$, there exist $t \le T_2$ such that $\|w_i^{(t)}\|_2 \ge 2\|w_i^{(T_2)}\|_2$, we have

$$
\begin{aligned}
|\langle w_i^{(T'_{i,1})}, \mathbf{M}_j\rangle|^2 &\ge \|w_i^{(T'_{i,1})}\|_2^2 - \sum_{j\in[d],\,j\notin\mathcal{N}_i} \langle w_i^{(T'_{i,1})}, \mathbf{M}_j\rangle^2 - \sum_{j\in[d_1]\setminus[d]} \langle w_i^{(T'_{i,1})}, \mathbf{M}_j^\perp\rangle^2 \\
&\ge 2\|w_i^{(T_1)}\|_2^2 - (1+o(1))\|w_i^{(T_1)}\|_2^2 \\
&\ge (1-o(1))\|w_i^{(T_1)}\|_2^2
\end{aligned}
\tag{85}
$$

which proves the claim.

At this substage, we have: If $i \in \mathcal{M}_j^*$, then from similar calculations as above, we can prove by induction that starting from $t = T'_{i,1}$, it holds:

$$
\begin{aligned}
|\langle w_i^{(t+1)}, \mathbf{M}_j\rangle| &\ge |\langle w_i^{(t)}, \mathbf{M}_j\rangle|\left(1+\Omega\left(\epsilon_j \frac{\eta \log\log d}{d}\right)\right) \\
&\ge \|w_i^{(t)}\|_2\left(1+\Omega\left(\epsilon_j \frac{\eta \log\log d}{d}\right)\right)
\end{aligned}
\tag{86}
$$

$$\sum_{j'\in[d],\,j'\neq j} \langle w_i^{(t+1)}, \mathbf{M}_{j'}\rangle^2 \le \sum_{j'\in[d],\,j'\neq j} \langle w_i^{(t)}, \mathbf{M}_{j'}\rangle^2 \left(1+\epsilon_j \frac{O(\eta)}{d\,\mathrm{polylog}(d)}\right)^2 \tag{87}$$

$$\sum_{j\in[d_1]\setminus[d]} \langle w_i^{(t+1)}, \mathbf{M}_j^\perp\rangle^2 \le \sum_{j\in[d_1]\setminus[d]} \langle w_i^{(t)}, \mathbf{M}_j^\perp\rangle^2 \left(1+\frac{O(\eta)}{d\,\mathrm{polylog}(d)}\right)^2 \tag{88}$$

which implies

$$
\begin{aligned}
|\langle w_i^{(t+1)}, \mathbf{M}_j\rangle| &\ge |\langle w_i^{(t)}, \mathbf{M}_j\rangle| \cdot \frac{\|w_i^{(t+1)}\|_2}{\|w_i^{(t)}\|_2} \\
&\ge (1-o(1))\|w_i^{(t+1)}\|_2
\end{aligned}
\tag{89}
$$

Theorem 3.2 (6) is proved. As for Theorem 3.2 (7), we can revisit case (c) from the three situations discussed earlier and then proceed by iteration. $\qquad\square$

# F    Theorem 3.3

At the final stage, we show that sparse activation of neurons naturally leads to convergence toward sparse solutions, thereby guaranteeing sparse representations. For all $t \geq T_2$:

**Lemma F.1.** *For all iterations $t$, the neurons $i \in [m]$ satisfy the following properties:*

*(a) For $j \in [d]$, if $i \in \mathcal{M}_j^\star$, then*

$$\left| \langle w_i^{(t)}, \mathbf{M}_j \rangle \right| \geq \Omega(1) \| w_i^{(t)} \|_2 \tag{90}$$

*(b) For $i \in [m]$, we have*

$$\| w_i^{(t)} \|_2 \leq O(1) \tag{91}$$

*(c) For each $j \in [d]$,*

$$\widehat{\mathfrak{F}}_j^{(t)} := \sum_{i \in \mathcal{M}_j} \langle w_i^{(t)}, \mathbf{M}_j \rangle^2 \leq O\left( \left( \frac{\epsilon_j}{\epsilon_{\max}} \right)^2 \tau \log^3 d \right) \tag{92}$$

*(d) Let $j \in [d]$ and $i \in \mathcal{M}_j^\star$, then there exists $C = \Theta(1)$ such that*

$$\left| \langle w_i^{(t)}, \mathbf{M}_j \rangle \right| \geq C \max_{i' \in \mathcal{M}_j} \left| \langle w_{i'}^{(t)}, \mathbf{M}_j \rangle \right| \tag{93}$$

*(e) For $i \notin \mathcal{M}_j$, it holds*

$$\left| \langle w_i^{(t)}, \mathbf{M}_j \rangle \right| \leq O\left( \frac{\epsilon_j}{\epsilon_{\max}} \frac{1}{\sqrt{d} \, \Xi_2^5} \right) \| w_i^{(t)} \|_2 \tag{94}$$

*(f) For any $i \in [m]$ and any $j \in [d_1] \setminus [d]$, it holds*

$$\left| \langle w_i^{(t)}, \mathbf{M}_j^\perp \rangle \right| \leq O\left( \frac{1}{\sqrt{d_1} \, \Xi_2^5} \right) \| w_i^{(t)} \|_2 \tag{95}$$

*(g) For all $i \in [m]$, the bias satisfies*

$$b_i^{(t)} \geq \frac{\mathrm{polylog}(d)}{\sqrt{d}} \| w_i^{(t)} \|_2 \tag{96}$$

**Definition F.1** (Optimal Learner). *We define a learner network that we deem as the "optimal" feature map for this task. Let $\kappa > 0$, we define $\theta^\star := \{\theta_i^\star\}_{i \in [m]}$ as follows:*

$$\theta_i^\star = \begin{cases} \dfrac{\sqrt{\tau}\, \kappa}{|\mathcal{M}_j^\star|} \mathbf{M}_j \cdot \mathrm{sign}\left( \langle w_i^{(T_2)}, \mathbf{M}_j \rangle \right), & \text{if } i \in \mathcal{M}_j^\star, \\ 0, & \text{if } i \notin \bigcup_{j \in [d]} \mathcal{M}_j^\star \end{cases} \tag{97}$$

*Furthermore, we define the optimal feature map $f_t^\star$ as follows. For $i \in [m]$, the $i$-th neuron of $f_{t,\theta}$ given weight $\theta_i \in \mathbb{R}^{d_1}$ is*

$$f_{t,\theta,i}(X_n) = \sum_{r=1}^L \left[ \left( \langle \theta_i, z_X^{(r)} \rangle - b_i \right) \mathbf{1}_{\langle w_i^{(t)}, z_X^{(r)} \rangle \geq b_i} - \left( -\langle \theta_i, z_X^{(r)} \rangle - b_i \right) \mathbf{1}_{-\langle w_i^{(t)}, z_X^{(r)} \rangle \geq b_i} \right]. \tag{98}$$

*Finally, we write $f_{t,\theta}$ as the concatenation*

$$f_{t,\theta}(\cdot) = \left( f_{t,\theta,1}(\cdot), \ldots, f_{t,\theta,m}(\cdot) \right)^\top \tag{99}$$

**Lemma F.2** (Optimality). *Let $\{\theta_i^\star\}_{i \in [m]}$ and $f_{t,\theta}$ be defined as in Definition F.1. When Lemma F.1, define the pseudo loss function*

$$\widetilde{\mathcal{L}}(f_{t,\theta^\star}, f_t) := \mathbb{E}\left[ -\tau \log\left( \frac{e^{\langle f_{t,\theta^\star}(X_n), f_t(X_{n'}) \rangle / \tau}}{\sum_{x \in \mathfrak{B}} e^{\langle f_{t,\theta^\star}(X_n), f_t(x) \rangle / \tau}} \right) \right] \tag{100}$$

*Then by choosing $\kappa = \Theta(\Xi_2)$, and assuming*

$$\sum_{i \in \mathcal{M}_j^\star} |\langle w_i^{(t)}, \mathbf{M}_j \rangle| \geq \Omega\left(\frac{\sqrt{\tau}}{\Xi_2}\right), \tag{101}$$

*we obtain the following loss guarantee:*

$$\widetilde{\mathcal{L}}(f_{t,\theta^\star}, f_t) \leq O\left(\frac{1}{\log d}\right) \tag{102}$$

*Proof of Theorem 3.3.* We start with the proof of convergence Theorem 3.3 (8).

Denote $w^{(t)} = (w_1^{(t)}, \ldots, w_m^{(t)})$, since our update is

$$w^{(t+1)} = w^{(t)} - \nabla_w \mathrm{Obj}(f_t) + \frac{1}{\mathrm{poly}(d_1)}, \tag{103}$$

we have

$$\eta \langle \nabla_w \mathrm{Obj}(f_t), w^{(t)} - \theta^\star \rangle = \frac{\eta^2}{2}\|\nabla_w \mathrm{Obj}(f_t)\|_F^2 + \frac{1}{2}\|w^{(t)} - \theta^\star\|_F^2 - \frac{1}{2}\|w^{(t+1)} - \theta^\star\|_F^2 + \frac{\eta^2}{\mathrm{poly}(d_1)}$$

$$\leq \eta^2 \mathrm{poly}(d) + \frac{1}{2}\|w^{(t)} - \theta^\star\|_F^2 - \frac{1}{2}\|w^{(t+1)} - \theta^\star\|_F^2 + \frac{\eta^2}{\mathrm{poly}(d_1)} \quad (104)$$

The proof of the above equation is as follows:

$$\langle x, y \rangle = \tfrac{1}{2}\left(\|x\|^2 + \|y\|^2 - \|x - y\|^2\right) \tag{105}$$

Let $x = a - c$, $y = a - b$, and substitute into the above equation.

$$\langle a - c, a - b \rangle = \tfrac{1}{2}\left(\|a - c\|^2 + \|a - b\|^2 - \|b - c\|^2\right) \tag{106}$$

Here we substitute the following three quantities into the three point identity:

$$a = w^{(t)}, \qquad b = \theta^\star, \qquad c = w^{(t+1)} = w^{(t)} - \eta \nabla_w \mathrm{Obj}(f_t) \pm \frac{\eta}{\mathrm{poly}(d_1)} \tag{107}$$

Thus, the original equation is proved. As for the inequality,

$$\|\nabla_w \mathrm{Obj}(f_t)\|_F^2 = \sum_{i=1}^m \|\nabla_{w_i} \mathrm{Obj}(f_t)\|^2 \tag{108}$$

Each term is $O(1)$, and since $m = \mathrm{poly}(d)$, the overall complexity is $\mathrm{poly}(d)$.

Now we will use the tools from online learning to obtain a loss guarantee: define a pseudo objective for parameter $\theta$

$$\begin{aligned}\widetilde{\mathrm{Obj}}_t(\theta) &:= \widetilde{\mathcal{L}}(f_{t,\theta}, f_t) + \tfrac{\lambda}{2} \sum_{i \in [m]} \|\theta_i\|_2^2 \\ &= \mathbb{E}\left[-\tau \log\left(\frac{e^{\langle f_{t,\theta}(X_n), f_t(X_{n'})\rangle/\tau}}{\sum_{x \in \mathfrak{B}} e^{\langle f_{t,\theta}(X_n), f_t(x)\rangle/\tau}}\right)\right] + \tfrac{\lambda}{2} \sum_{i \in [m]} \|\theta_i\|_2^2\end{aligned} \tag{109}$$

Which is a convex function over $\theta$ since it is linear in $\theta$ (for a fixed $f_t$, we can consider $\widetilde{\mathcal{L}}(f_{t,\theta}, f_t)$ to be convex with respect to $\theta$, because $f_{t,\theta}(x)$ is linear, and softmax + log is a convex composition; the regularization term is convex).

Moreover, we have

$$\widetilde{\mathrm{Obj}}_t(w^{(t)}) = \mathrm{Obj}(f_t), \tag{110}$$

and

$$\nabla_{\theta_i} \widetilde{\mathrm{Obj}}_t(w_i^{(t)}) = \nabla_{w_i} \mathrm{Obj}(f_t) \tag{111}$$

Thus we have

$$\eta\langle\nabla_w\mathrm{Obj}(f_t), w^{(t)} - \theta^\star\rangle = \eta\langle\nabla_\theta\widetilde{\mathrm{Obj}}_t(w^{(t)}), w^{(t)} - \theta^\star\rangle$$

$$\overset{(1)}{\geq} \widetilde{\mathrm{Obj}}_t(w^{(t)}) - \widetilde{\mathrm{Obj}}_t(\theta^\star)$$

$$\geq \widetilde{\mathrm{Obj}}_t(w^{(t)}) - \mathbb{E}\left[-\tau\log\left(\frac{e^{\langle f_{t,\theta^\star}(X_n), f_t(X_{n'})\rangle/\tau}}{\sum_{x\in\mathfrak{B}} e^{\langle f_{t,\theta^\star}(X_n), f_t(x)\rangle/\tau}}\right)\right] - \frac{\lambda}{2}\sum_{i\in[m]}\|\theta_i^\star\|_2^2$$

$$\overset{(2)}{\geq} \widetilde{\mathrm{Obj}}_t(w^{(t)}) - O\left(\tfrac{1}{\log d}\right) - \sum_{i\in[m]} O(\lambda\|\theta_i^\star\|_2^2)$$

$$\geq \mathrm{Obj}(f_t) - O\left(\tfrac{1}{\log d}\right) \tag{112}$$

(1) is because the surrogate objective function $\widetilde{\mathrm{Obj}}_t$ is a convex function with respect to $\theta$, so we can use a first-order convex lower bound: $f(\theta) - f(\theta') \leq \langle\nabla f(\theta), \theta - \theta'\rangle$. (2) is because $\sum_{i\in[m]}\lambda\|\theta_i^\star\|_2^2 = \sum_{j\in[d]}\sum_{i\in\mathcal{M}_j^\star}\lambda\|\theta_i^\star\|_2^2 = \sum_{j\in[d]}\sum_{i\in\mathcal{M}_j^\star}\lambda\frac{\tau\kappa^2}{|\mathcal{M}_j^\star|^2} = \sum_{j\in[d]}\lambda\frac{\tau\kappa^2}{|\mathcal{M}_j^\star|} = \frac{\lambda\tau\kappa^2}{|\mathcal{M}_j^\star|}$

Now choosing $\kappa = \Theta(\Xi_2) \leq \frac{1}{\lambda d}$ (so that $\sum_{i\in[m]}\lambda\|\theta_i^\star\|_2^2 < \frac{1}{\log d}$), and by a telescoping summation, we have

$$\frac{1}{T}\sum_{t=T_3}^{T_3+T-1}\left(\mathrm{Obj}(f_t) - O\left(\tfrac{1}{\log d}\right)\right) \leq \frac{1}{T}\sum_{t=T_3}^{T_3+T-1}\eta\langle\nabla_w\mathrm{Obj}(f_t), w^{(t)} - \theta^\star\rangle$$

$$\leq \frac{O(\|w^{(T_3)} - \theta^\star\|_F^2)}{T\eta}$$

$$= \frac{O\left(\|w^{(T_3)}\|_F^2 + \|\theta^\star\|_F^2 - 2\mathrm{Tr}((w^{(T_3)})^\top\theta^\star)\right)}{T\eta}$$

$$\leq \frac{O\left(\|w^{(T_3)}\|_F^2 + \|\theta^\star\|_F^2\right)}{T\eta}$$

$$\leq \frac{O\left(m\|w_i^{(T_3)}\|_2^2\right)}{T\eta}$$

$$\leq O\left(\tfrac{m\Xi_2}{T\eta}\right) \tag{113}$$

Since $T\eta \geq m\Xi_2^{10}$, this proves the claim.

For Theorem 3.3 (9), we have

$$w_i^{(t)} = \sum_{j\in\mathcal{N}_i, j\in[d]}\langle w_i^{(t)}, \mathbf{M}_j\rangle\mathbf{M}_j + \sum_{j\notin\mathcal{N}_i, j\in[d]}\langle w_i^{(t)}, \mathbf{M}_j\rangle\mathbf{M}_j + \sum_{j\in[d_1]\setminus[d]}\langle w_i^{(t)}, \mathbf{M}_j^\perp\rangle\mathbf{M}_j^\perp$$

$$\leq \sum_{j\in\mathcal{N}_i, j\in[d]}\langle w_i^{(t)}, \mathbf{M}_j\rangle\mathbf{M}_j + \sum_{j\notin\mathcal{N}_i, j\in[d]}O\left(\frac{\epsilon_j}{\epsilon_{\max}}\frac{\|w_i^{(t)}\|_2}{\sqrt{d}\,\Xi_2^5}\right)\mathbf{M}_j + \sum_{j\in[d_1]\setminus[d]}O\left(\frac{\|w_i^{(t)}\|_2}{\sqrt{d_1}\,\Xi_2^5}\right)\mathbf{M}_j^\perp$$

$$= \sum_{j\in\mathcal{N}_i, j\in[d]}\alpha_{i,j}\mathbf{M}_j + \sum_{j\notin\mathcal{N}_i, j\in[d]}\alpha'_{i,j}\mathbf{M}_j + \sum_{j\in[d_1]\setminus[d]}\beta_{i,j}\mathbf{M}_j^\perp \tag{114}$$

Under the condition of $|\mathcal{M}_j| \neq 0$ (if $|\mathcal{M}_j| = 0$, then it is not a target within $\mathcal{N}_i$, and thus it becomes meaningless), and due to Lemma F.1: The proof is complete. For each feature $\mathbf{M}_j$, there are at most $o(m/d)$ many $i \in [m]$ such that $j \in \mathcal{N}_i$: It follows from the proof of Lemma C.1 that $\mathbb{P}[i \in \mathcal{M}_j] = \frac{1}{d^{\Omega(1)}}$, and at least $\Omega(d^{\omega_1})$ many $i \in [m]$ such that $\mathcal{N}_i = \{j\}$: From Lemma C.1, we recall that $|\mathcal{M}_j^\star| \geq \Omega(d^{\omega_1})$ If a neuron belongs to $\mathcal{M}_j^\star$, then it must not belong to $\mathcal{M}_{j'}$

$\square$