# OpenReview forum: "On the Training Dynamics of Contrastive Learning with Imbalanced Feature Distributions: A Theoretical Study of Feature Learning"
_NeurIPS.cc/2025/Workshop/UniReps — UniReps2025_

### Official Review · Reviewer_zggc · 2025-09-03

**Confidence:** 4

**Review:**

**Summary:**

The authors study the learning dynamics of contrastive learning when applied on imbalanced data, i.e. when data presents classes that appear very frequently in the data (majority classes) and classes that appear very infrequently (minority classes). The analysis is carried out for a small transformer model consisting of one attention layer and one MLP layer, where only the MLP weights are learned (the attention weights are frozen). The theoretical and empirical analysis suggest that the majority features are represented by multiple neurons, unlike the minority features.

**Review:**

The introduction is well written and provides a good motivation for the rest of the paper. Overall the topic got me intrigued and is a good fit for this workshop.

That being said, the description of technical results lacks in clarity. I suspect this might have been caused by the authors compressing a lengthier version of the work to only four pages. I found multiple cases where symbols are simply not properly defined, which unfortunately rendered the technical results impossible to understand, at least for me. Maybe someone with the right a priori knowledge about this type of analysis could follow. But still, I believe someone having the right level of mathematical knowledge (which I think have) should be able to extract the essence of the technical results with some effort just by reading the main text. My suggestion would be to cut the abstract and the intro and spend more time explaining the various objects necessary to understand the theorems. Below I list some examples of things that got me confused:
- Equation (3): The index t is not explained here. Is it a typo and so should we replace t by r here? Or is it the time index of the optimization algorithm? Which optimization algorithm?
- Theorem 3.1: M_j is never properly defined. Similarly for w_i^(T). It is the result of making T steps of SGD?
- Theorem 3.2: \mathcal{M}*_j is never properly defined. Again, this makes the statement impossible to understand.

**Minor point:**

Equation (1): Is this stopGrad standard?

**Score:**

3

**Topic Fit:**

3

---

### Official Review · Reviewer_8ZqU · 2025-09-14
**Limited clarity and insufficient experimental verification cast doubt on the novel contributions of the paper**

**Confidence:** 3

**Review:**

This paper studies the training dynamics of a transformer-based encoder model that is trained with a contrastive loss on imbalanced data. The authors show that, during training, the weights of the last-layer MLP neurons develop alignments along different features of the data depending on their frequencies. (The data is generated from a sparse coding model, in which positive pairs share the same group of features.) They establish corresponding theoretical bounds at three stages of training. While the problem of understanding the features learned in contrastive learning and their association with data statistics is in general interesting and important, the new contributions of this paper are not clear to me.

The theorems are presented without much motivation and follow up discussion on their significance. Importantly, the experimental data from simulations are limited showing only one snapshot of training in which more neurons are aligned with “majority features” – how surprising is this result, and has this been studied previously in the literature? Further, the timing of the three stages of learning and the associated bounds are not verified in the experiments.

In the abstract, it is mentioned that:  “We further show that minority features diminish neurons’ representational capacity, increase the need for more complex architectures, and impair the separation of ground-truth features from noise”. This conclusion is not immediately evident from the main text and requires further explanation.

Some minor comments:

- In the main text, the data generation process is not described specifically, and some key terms such as majority and minority features are not defined. It would help to define these more conretely in the main text.
- The times T1, T2 are not defined in the main text, though they are used in the theorems.
- $C_z$ in Eq. 4 is not defined.
- $M_j^*$ in line 99 is not defined.
- Experimental details are missing. For example, how are the fixed weights of the attention layer set? How are the weights of the MLP layer initialized? Do the findings depend on these choices?
- In Figure 1, the title suggests the values are squared of cosine, but the legend text suggests cosine. Which one is correct? Additionally, the caption says 10 features shown but it appears that one column is missing from the figure.

**Score:**

2

**Topic Fit:**

1

---

### Official Review · Reviewer_F5wu · 2025-09-14
**The paper demonstrates how the alignment of weights with features evolve during training for contrastive learning with imbalanced data.**

**Confidence:** 3

**Review:**

This paper by adopting the sparse coding as the data model, rigorously identifies three stages of how neurons align with the dictionary atoms during the training under gradient descent. Understanding the precise dependence of alignment between the weight vectors and the features will help to alleviate the effect of imbalanced data in contrastive learning. Although the problem setting and the results are important, further work is required to improve the clarity and readability of the paper. Below I will go over some of the mentioned points.

**Strengths**:

1. Provides a theoretical framework for understanding the dynamics of feature learning in contrastive learning.
3. The bounds provided in the results of the theorems are intuitive and easy to follow.
2. The architecture considered (Attention + MLP) is highly relevant in practice.


**Weaknesses and comments**:

1. Missing definitions and notations:

    1.1 The mathematical definition of gradient descent update is missing. What assumptions are made about the learning rate?

    1.2 $Obj$ and $\hat{Obj}$ are not defined.

    1.3. The frequency $\epsilon_j$ is undefined.

    1.4 The definitions of $T_1$ and $T_2$ are missing from Theorem 3.1, 3.2.

    1.5 In the numerical experiment section, the details of the synthetic data are missing. How are the dictionary atoms $M_j$ generated?

2. About the assumptions and the theoretical results:

      2.1 Why are the attention weights taken to be identity and not Gaussian? Usually such models are initialized with normal or truncated normal distribution. Furthermore,if the weights were to be taken normal, the model will more closely resemble a random features model, which has been a common subject of study. [*]

      2.2 Does Theorem 3.1 hold for every $i \in [m]$? In Theorem 3.2, what happens for $i \notin M_j^\ast$?

3. I highly recommend performing experiments on real-world data to corroborate the qualitiative message about the three stages of training.

4. On a more tangential note, it would be interesting to examine how the validation/test loss corresponds to the three proposed stages of training, for instance by plotting the test loss against training steps.

[*] Fu, Hengyu, et al. "What can a single attention layer learn? a study through the random features lens." Advances in Neural Information Processing Systems 36 (2023): 11912-11951.

**Score:**

4

**Topic Fit:**

3

---

### Official Review · Reviewer_yvon · 2025-09-16
**Theoretical contribution is meaningful, but the simplified setting limits immediate applicability**

**Confidence:** 3

**Review:**

This paper provides a theoretical analysis of contrastive learning dynamics under imbalanced feature distributions, using a simplified Transformer-MLP model with sparse coding. It identifies three training stages where majority features are learned faster than minority features, and shows that imbalance reduces feature purity, increases architectural complexity, and hinders noise separation.

**Strengths**:

Rigorous theoretical framework for imbalanced contrastive learning.

Clear characterization of three-stage training dynamics.

Novel insights into minority feature suppression and noise interactions.


**Weaknesses**:

Highly simplified setting (single-head attention, fixed identity matrices, synthetic data).

No empirical validation on real-world datasets or architectures.

Limited discussion of practical mitigation strategies for imbalance.

Assumptions (e.g., sparse features, identity attention) may not hold in practice.

**Final assessement**:
I would recommend weak accept for this paper. The theoretical contribution is meaningful and offers new insights into imbalance in contrastive learning, but the simplified setting and lack of empirical validation may limit immediate applicability. The work would be strengthened by experiments on real data and extensions to standard architectures like multi-head Transformers.

**Score:**

3

**Topic Fit:**

2

---

### Official Review · Reviewer_iT4p · 2025-09-19
**Review of 'On the Training Dynamics of Contrastive Learning with Imbalanced Feature Distributions'**

**Confidence:** 4

**Review:**

Summary:

The study aims to provide a theoretical understanding of how imbalanced data affects representation learning and leads to bias in the context of contrastive learning. Through the proposed framework, the authors analyzed the dynamics of Transformer-based encoders under imbalanced data and revealed that model training can be divided into three stages, each with distinct properties.

The authors included proofs for their theoretical analysis, which helped outline the patterns of neuron weight updates across the three stages. This establishes a solid analytical foundation for the work.

They also validated their findings using synthetic data, demonstrating how majority and minority features are learned differently, and discussed the implications of imbalanced data on model performance and representation quality.

Potential areas for improvement:

- Sparse coding was used out of theoretical necessity, and while it simplifies the analysis, it may not fully capture the complexity of feature learning in scenarios where features exhibit more complex relationships.
- What could be some alternative explanations for the results observed in the numerical experiments?
- Beyond defining ordinary and lucky neurons, could there also be "unlucky neurons" that initially align very poorly with features or/and fail to specialize during training?
- Would the type and properties of noise affect the conclusions of the study?

**Score:**

4

**Topic Fit:**

2